# The Association between Postpartum Practice and Chinese Postpartum Depression: Identification of a Postpartum Depression-Related Dietary Pattern

**DOI:** 10.3390/nu14040903

**Published:** 2022-02-21

**Authors:** Ai Zhao, Shanshan Huo, Yuefeng Tan, Yucheng Yang, Ignatius Man-Yau Szeto, Yumei Zhang, Hanglian Lan

**Affiliations:** 1Vanke School of Public Health, Tsinghua University, Beijing 100084, China; aizhao18@tsinghua.edu.cn (A.Z.); shanshanhuo_pku@163.com (S.H.); tany1995@163.com (Y.T.); yyucheng@mail.tsinghua.edu.cn (Y.Y.); 2Institute for Healthy China, Tsinghua University, Beijing 100084, China; 3Inner Mongolia Dairy Technology Research Institute Co., Ltd., Hohhot 010110, China; szeto@yili.com; 4Yili Innovation Center, Inner Mongolia Yili Industrial Group Co., Ltd., Hohhot 010110, China; 5Department of Nutrition and Food Hygiene, School of Public Health, Peking University, Beijing 100191, China

**Keywords:** postpartum practice, postpartum depression, dietary behaviors, dietary pattern

## Abstract

Conflicting results of Chinese traditional postpartum practices have aroused concerns over their health effects. The role of postpartum practices in postpartum depression (PPD) is still a matter of discussion, especially from a dietary practice point of view. The current study was designed to (1) explore the association between postpartum practices and PPD, (2) to identify the dietary pattern related to PPD, and (3) to identify the possible pathways among postpartum practices and PPD. This study is part of the YI Study, which was a cross-sectional study conducted in 10 cities in China. Data for 955 postpartum women were used in the current analysis. The Edinburgh Postnatal Depression Scale (EPDS-10) was used to evaluate PPD with a cutoff value of 9. Postpartum practice was based on the participants’ self-reported practices. Individual practices were recorded and categorized as dietary and non-dietary practices. The dietary pattern was identified based on the food intake frequencies of 25 food groups using the method of reduced rank regression. Structural equation modeling was used to explore the potential pathways between postpartum practices and PPD. The current study observed significant associations between postpartum practices and PPD (Adjusted OR = 1.41, 95% CI: 1.04–1.90). A similar trend was also found between dietary postpartum practices and PPD (Adjusted OR = 1.39, 95%CI: 1.03–1.88) but not for non-dietary practices and PPD (Adjusted OR = 1.26, 95%CI: 0.92–1.75). A PPD-related dietary pattern was identified with the characteristics of a high intake of meat and eggs and a lower intake of vegetables, mushrooms, and nuts. This dietary pattern was significantly associated with a higher chance of adhering to postpartum practice (Adjusted OR = 1.26, 95% CI: 1.10–1.44). Based on the pathway analysis, this study also observed the association between postpartum practices and PPD, and the association between dietary practices and PPD were both mediated by sleep quality. In conclusion, this study demonstrated that a substantial proportion of women in modern China experience traditional postpartum Chinese practices and that either overall or dietary-related postpartum practices are associated with a higher risk of PPD. The current dietary practices in postpartum rituals may play an important role in developing PPD. A culturally embedded, science-based dietary guideline is required to help women to achieve both physical and psychological health in the postpartum period.

## 1. Introduction

Care in the postpartum period is critical, as many physiological and psychological changes occur that pose great risks to short- and long-term maternal health [1]. A unique lifestyle and confinement diet, which is considered beneficial for postpartum women, is traditionally observed in China; however, its health effects have recently aroused great controversy [2].

This traditional postpartum practice includes a series of taboos, such as avoiding showering, not drinking or touching cold water, and the limited consumption of fruits and vegetables. Meanwhile, it encourages increasing the intake of certain foods, such as chicken soup and animal viscera [3,4,5,6]. Commonly, these practices would be adopted in the first 1 to 2 months postpartum, but the ideas and certain practices would last throughout the entire lactation period [2]. In our previous study, we observed that such dietary practices could bring about higher iron intake for lactating mothers, but they could also bring a potential risk of vitamin deficiency by lacking fresh vegetables and fruits [7]. The role of postpartum practices raises concerns over not only physical but also mental health. One case study reported the therapeutic effects of Chinese traditional postpartum practices on postpartum depression (PPD) by strengthening a mother’s self-esteem, providing emotional support, and buffering the stress experienced in early motherhood [8]. This finding was also observed in one cross-sectional study conducted in Hunan province, China, which revealed that low and moderate adherence to postpartum rituals was associated with a higher risk of PPD [9]. By contrast, some other studies conducted in various populations demonstrated the opposite result [10]. It is worthy of note that the roles of individual postpartum practices may vary widely. One Taiwanese study evaluated the different effects of certain practices and found that behaviors including avoiding bathing and showering were associated with a low depression status; however, behaviors including avoiding touching cold water and squatting contributed to a high depression status [11]. Apparently, limited studies yield a mixed result, and studies focusing on the psychological effects of postpartum dietary practices are sparse.

Many foods and nutrients have a mood regulatory function [12]. For instance, it has been well documented that omega-3 fatty acids can be used in PPD prevention and treatment [13]. Fiber was also found to help decrease maternal depression by alleviating obesity [14]. One Spanish study reported that postpartum women were more optimistic when they adhered to a healthy diet [15]. In addition to the biological role of traditional postpartum rituals, for women whose previous dietary habits were not aligned with them, such rituals may bring social pressure.

Globally, 10–20% of women are affected by PPD during the first year after childbirth, and this prevalence is around 16% in China, making it a significant public health concern [16,17]. Therefore, whether women could obtain psychological benefits from postpartum practices is an interesting topic to explore. This study was designed to investigate the association between postpartum practices and PPD, especially from a dietary perspective.

## 2. Materials and Methods

This study was part of the Young Investigation (YI Study). The YI Study was a cross-sectional study conducted from 2019 to 2020, which focused on maternal and child nutrition and health. In the present study, with a purposive sampling approach, lactating women were selected from 10 cities in China (Beijing, Chengdu, Guangzhou, Hohhot, Lanzhou, Nanchang, Ningbo, Shenyang, Suzhou, and Xuchang) according to their economic level and geographic location. In each city, one hospital or one community-based maternal and child healthcare center was selected for convenience. Then, according to lactating women’ visiting time, women were recruited in each study site until the designed sample size (at least 90 individuals in each city) was satisfied. The inclusion criteria for participants were (1) healthy women who were in the first 0–365 days postpartum, (2) aged between 20 and 45 years, (3) with a healthy full-term delivery, (4) without smoking and alcohol abuse, (5) without mastitis and any infectious diseases on the day of the investigation, and (6) without cardiovascular diseases and severe metabolic diseases. Women with significant memory loss were excluded. A total of 975 women completed the survey. Finally, data from 955 women were included in the current analysis (20 women were excluded because key values were missing in the database, including lactation stage, depressive symptom assessment, and whether they adhered to the postpartum customs).

### 2.1. Data Collection

An interviewer-administered questionnaire was used to collect the data. Training of the interviewers, the initial site survey, and preliminary questionnaire testing were completed prior to data collection. 

Maternal postpartum depressive symptoms were evaluated with the Chinese version of the 10-item Edinburgh Postnatal Depression Scale (EPDS-10) [18]. The scale consists of 10 short statements. The participants checked off one of four possible answers that was closest to how she had felt during the past week. The total score on the EPDS-10 could range from 0 to 30. As Huang and colleagues suggested, for the Chinese population, a total score ≥9 could be considered as mild depression, while ≥13 could be considered as severe depression [19].

Whether women adhered to certain postpartum practices or not was assessed by one question: “Do you adhere to any taboos, rituals, and prescriptions in the postpartum period?” For women who answered “yes,” an open-ended question “Please describe the specific practice you adhere to” was followed. All the information reported by the women was documented verbatim. The words/descriptions with similar meanings were combined by two independent researchers. Inconsistent descriptions were discussed and achieved agreement with the help of the third researcher. Finally, similar words/descriptions were given a unified expression and categorized as dietary practices and non-dietary practices, respectively. Finally, the unified expression of practices was used to draw word clouds with the “stylecloud” package in PyCharm Community Edition (3 March 2020).

For dietary intake, the intake frequencies of 25 food groups in the past month based on a semi-quantitative Food Frequency Questionnaire were used in the current analysis (including refined grain, coarse grain, tubers, dark green vegetables, red-orange vegetables, light colored vegetables, other fresh vegetables, pickled vegetables, mushrooms, fungi, seaweed, soybeans, nuts, fruit, animal meat, poultry meat, liver, other offal, processed meat, seafood, freshwater fish, egg, dairy products, soup, and snacks). All the foods involved in the current analysis were selected according to the Dietary Guidelines for Chinese lactating women and common postpartum practices reported by previous studies [3,7,20]. A PPD-related dietary pattern was constructed using reduced rank regression (RRR) analysis with the intake frequencies of 25 food groups as input variables and the EPDS-10 score as the response variable. The PROC PLS statement in SAS (SAS Institute, Cary, NC, USA) was used to conduct RRR analysis. The significance level of the variables was set to 0.1.

Sociodemographic characteristics were investigated with a self-designed questionnaire. The International Physical Activity Questionnaire-Short Form (IPAQ-SF) was used to assess the physical activity level [21]. The total physical activity level was measured by weighted metabolic equivalent hours per week (MET·hours/week), which was calculated as the sum of duration × frequency per week × MET intensity per sport [22]. Sleep quality was assessed based on self-report with a question of “Please rate your sleep quality in the recent 1 week” that was used to assess sleep quality as “very good = 1, good = 2, bad = 3, or very bad = 4”. Weight and height were measured in the field, and body mass index (BMI) was calculated. According to the Chinese BMI standard, BMI < 18.5, 18.5–23.9, 24–27.9, and ≥28 kg/m^2^ were defined as underweight, normal weight, overweight, and obese, respectively. 

### 2.2. Statistics

Data were analyzed with SAS version 9.4 (SAS Institute, Cary, NC, USA). Data are presented as means ± standard deviation (SD) or percentages. The differences in sociodemographic characteristics, PPD, and PPD-related dietary patterns between women with certain traditional postpartum practices and those without were tested with the chi-square test or independent t-test. A binary regression model was used to explore the associations of PPD and PPD-related dietary patterns with postpartum practice. Multivariable logistic regression was also conducted to examine the above associations after adjusting for potential confounders (maternal age, family income, lactation stages, numbers of family members, and self-reported sleep quality). All *p* values were two-sided, and statistical significance was defined as *p* < 0.05. Structural equation modeling (SEM) was established to find the possible pathway for postpartum practices and PPD using a weighted least-squares mean- and variance-adjusted (WLSMV) estimator. Mplus Version 7.4 (Muthén & Muthén Inc., Los Angeles, CA, USA) software was applied for SEM analysis.

## 3. Results

### 3.1. Prevalence of PPD

In the studied population, 33.7% of the participants were screened as having PPD (EPDS ≥ 9). The prevalence of PPD among people with different characteristics is shown in Table 1. Women older than 30 years, those with a relatively high family income, and those who self-reported having poor sleep quality were more likely to have PPD.

### 3.2. Association between PPD and Postpartum Practices

In the studied population, 59.5% of participants self-reported adhering to certain postpartum practices, with 28.5% of women self-reporting having non-dietary practices and 36.3% of them reporting having certain practices related to their diet. The practices self-reported by participants are shown in Figure 1.

Women’s adherence to overall postpartum practices and dietary practices were associated with a higher risk of PPD (Table 2). No association was observed between non-dietary practices and PPD.

### 3.3. PPD-Related Dietary Pattern

A PPD-related dietary pattern was derived using the RRR method with PPD as a response variable. Therefore, only one dietary pattern was identified. The PPD-related dietary pattern was characterized by high loadings of meat and eggs and a low intake of vegetables, mushrooms, and nuts (see Figure 2).

Compared with women who did not adhere to postpartum practices, women who experienced these practices had a significant PPD-related dietary pattern loading (Table 3).

### 3.4. Pathway Analysis

Analysis of the pathway from postpartum practices to PPD is shown in Figure 3. The final SEM model indicated a good fit: root mean square error of approximation (RMSEA) = 0.035, comparative fit index (CFI) = 0.955, Tucker–Lewis index (TFI) = 0.821, and weighted root mean square residual (WRMR) = 0.595. The specific standardized effect of PPD showed that adherence to postpartum practices and having poor self-reported sleep quality had significant positive associations with PPD. Meanwhile, older age was negatively associated with maternal PPD. Adherence to practices was also directly associated with poor sleep quality. The association between postpartum practices and PPD was mediated by sleep quality; the pathway coefficient value of the total effect was 0.142, in which the direct effect of practices was 0.122 and the indirect effect was 0.021. Analysis of the pathway from postpartum dietary practices to PPD was similar to the above and is shown in Appendix A.

## 4. Discussion

The current study demonstrates that a substantial proportion of women in modern China follow the traditional Chinese postpartum practices intended to maintain body harmony. However, a significant positive association between postpartum practices and the risk of PPD was observed, especially for those dietary practices. To the best of our knowledge, this is the first study to identify a PPD-related dietary pattern, and this pattern is in line with the dietary characteristics in postpartum dietary practices. 

### 4.1. Association of Postpartum Practices with PPD

It is well known that individual health behavior is embedded in cultural patterns. Although traditional postpartum practices, which were considered to be able to help rebuild the balance within the female body according to the cosmic dual principle of yin and yang in traditional Chinese medicine, have influenced women’s behavior for thousands of years [23], currently, an increasing number of studies have brought forth questions as to the health benefits of traditional postpartum practices. Regarding mental health, previous studies have addressed both the health benefits and the detriments of Chinese postpartum practices and have reached inconsistent conclusions [24,25,26]. Our study revealed that postpartum practices are associated with a higher risk of PPD. 

PPD is one of the most common complications during the perinatal period, and its risk factors involve physical, psychological, and social aspects. Some studies reporting on the beneficial effects of postpartum practice mentioned that women who adhere to the traditional practice usually obtain more family support and usually have a better relationship with family members [27,28]. By contrast, non-adherence to the traditional postpartum rituals was considered as obeying the social norm, which brings extra pressure during a period of heightened susceptibility to depression occurrence [29]. On the other hand, postpartum practices possibly contain elements hazardous to postpartum psychological health, especially when women are not satisfied with the traditional practices. One study revealed that fewer than 20% of women expressed complete satisfaction toward their own postpartum experiences [9]. Another study in Beijing, China, reported that around 50% of women perceived the postpartum rituals as useless [26]. Unfortunately, women’s satisfaction with postpartum practice in this study was not assessed.

In addition, based on the structural equation model developed in the current study, we found a strong mediation effect of sleep quality towards the association between postpartum behaviors and PPD. Some previous studies once reported that less sleep or poor quality sleep and adherence to customs were all associated with PPD, but the pathways of these factors were not examined [10,11]. It is well documented that poor quality sleep triggers PPD [30]. However, studies on the association between postpartum practice and sleep are rare. It is reasonable to infer that some postpartum practices may greatly impact on women’s sleep, such as the discomfort caused by avoiding showering, lack of physical activity, and frequent urination at night from drinking too much soup. In addition to sleep, other studies propose that women’s muscle endurance is another mediator between postpartum practice and PPD [31]. Unfortunately, muscle endurance was not measured in this study and only self-reported sleep quality was used in the current analysis. Longitudinal studies are still required to examine the pathway between postpartum practice and mental health.

### 4.2. PPD-Related Dietary Pattern

It is well documented that many nutrients can contribute to mental health. For PPD, previous studies revealed that adherence to a healthy diet, sufficient intake of fiber, and supplementation with polyunsaturated fatty acids could be beneficial in reducing the risk of PPD; meanwhile, iron deficiency may be associated with a higher risk of adverse mental consequences [14,15,32,33].

Dietary-related postpartum practice is one of the most common rituals followed by women; however, whether traditional Chinese dietary practices could impact on mental health has not yet been examined. With a reduced rank regression approach, we identified a PPD-related dietary pattern, with the characteristics of high intake of meat and eggs and low intake of vegetables, mushrooms, and nuts. Apparently, this pattern is similar to the common postpartum Chinese dietary rituals, and we further confirmed that women who adhere to postpartum practice had a significantly higher factor loading for this pattern. Regarding each food group, previous studies revealed that the association between meat and depression might be represented as a “U” shape. Although one meta-analysis showed a protective effect of moderate meat intake associated with lower depression and anxiety compared with meat abstention [34], excessive intake of meat, especially red meat, was associated with a higher risk of depression [35]. Meanwhile, a previous study revealed that not consuming three servings of nuts a week and not eating two servings of vegetables a day are predictors of depression, which supports our data [35]. Therefore, it is reasonable to conclude that dietary practices in the postpartum period are independently associated with a higher risk of PPD.

In addition, one study reported that dietary patterns could interact with sleep in depression [36]. A similar finding was also observed in the current study. This result could be explained by the existence of certain key nutrients involved in both sleep and mental regulation, such as polyunsaturated fatty acids, vitamin C, and vitamin B [37,38]. Calorie-rich food is another common risk factor for both sleep and depression [39,40,41]. It should be highlighted that the current postpartum dietary practice did not completely fit the current evidence-based dietary guidelines and could therefore lead to an insufficient intake of polyunsaturated fatty acids, vitamin C, and vitamin B and an excessive intake of energy, which are the mental health-related dietary components mentioned above [20].

### 4.3. Limitations

The strength of this study is that we explored the association between postpartum practice and depression with a sample from different regions of China. To the best of our knowledge, this is the first study to identify a PPD-related dietary pattern. However, the inherent limitations of this study were unavoidable because of its cross-sectional design. First, causality could not be examined, and emotional lability could also influence dietary intake. Second, regarding the methodology, the sampling method was not randomized, so a selection bias may exist. For example, all of the participants were recruited from urban areas, and we infer that traditional postpartum practice may be more common in rural areas. The non-randomized sampling method and a relatively small sample size may also undermine the reliability of the current results. The third concern is the measurement of the variables. In this study, both the food intake frequencies and intake amount were collected with Food Intake Frequencies Questionnaire (intake in the past 1 month) and one time of 24-hour dietary recall; however, considering that the representativeness of a relatively long term of dietary habit, reducing the recall bias and reducing the misclassification by inaccurately estimating the food intake amount, only the intake frequencies from Food Intake Frequencies Questionnaire were used to identify the PPD-related dietary pattern. However, the recall bias, the classification bias, and the confounding bias may still exist, and the intake amount might also be a characteristic of a certain dietary pattern. A 3-day 24-hour dietary record is recommended to further identify the dietary pattern. The sleep quality in current study was assessed according to the participants’ self-report. Although in this study, we focused more on the subjective feeling of participants. Because of the fact that this study revealed a crucial role of sleep quality in developing depression, a more detailed measurement of sleep (including sleep hours and quality) measured with a validated questionnaire or a more accurate tool is highly needed. Finally, some of the potential confounders were not fully collected and controlled, such as the maternal history of mental diseases and the current health status of infants. In addition, unmeasurable confounders may exist, and the results in current study should be treated with caution. Further prospective studies with a larger sample size are needed to examine the effects of postpartum practices on women’s health, especially from a dietary perspective.

## 5. Conclusions

The current study revealed an association between postpartum practice and PPD. The dietary pattern, characterized as a high intake of meat and eggs and a lower intake of vegetables, mushrooms, and nuts, is identified as being highly related to PPD. Although Chinese postpartum rituals were supposed to bring a certain benefit to women’s recovery, the current dietary postpartum practice in China is different from the dietary recommendations, and its health effects on women require serious evaluation. A better understanding of cultural practices and modern knowledge based on science could help healthcare professionals provide culturally sensitive care to meet both the physical and psychological needs of postpartum women.

## Figures and Tables

**Figure 1 nutrients-14-00903-f001:**
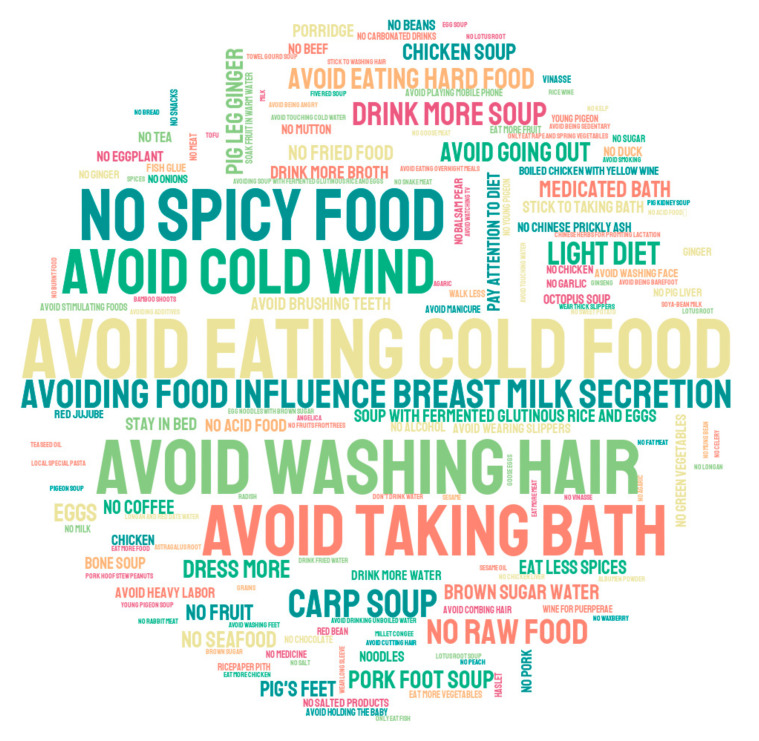
The traditional postpartum practices reported by lactating women in 10 cities of China. A larger word size indicates a higher reported frequency.

**Figure 2 nutrients-14-00903-f002:**
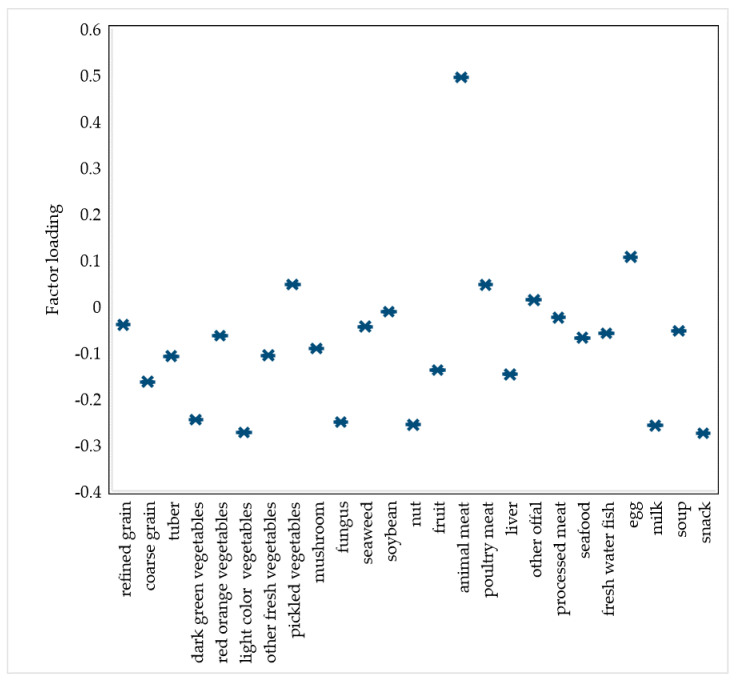
The factor loading of each food item in the postpartum depression-related dietary pattern. Reduced rank regression (RRR) analysis was conducted with the intake frequencies of 21 food groups as input variables and the EPDS-10 score as the response variable.

**Figure 3 nutrients-14-00903-f003:**
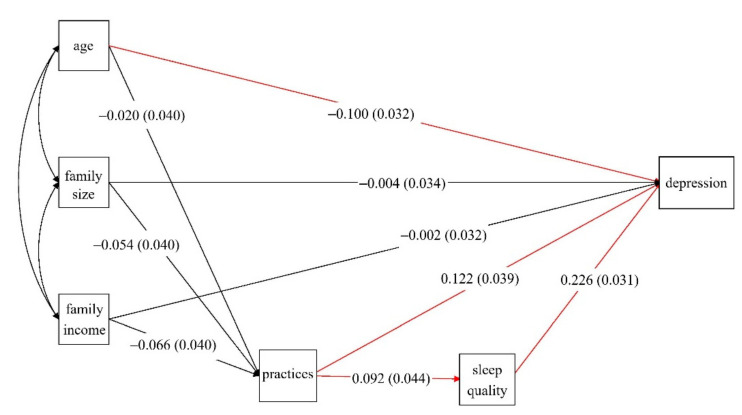
The pathways among postpartum practices and postpartum depression. The standardized effects (*p* values) are presented, and the red lines indicate the associations with significance.

**Table 1 nutrients-14-00903-t001:** Postpartum depression among women with different sociodemographic characteristics (*N*, %).

	Postpartum Depression		*p*
	Yes	No	
Lactation stages			0.126
15–42 days postpartum	161 (77.0)	42 (23.0)	
>42 days postpartum	535 (71.7)	211 (28.3)	
Age (years)			0.045
<30	601 (71.8)	236 (28.2)	
≥30	92 (80.7)	221 (19.3)	
Education experience			0.276
Senior school or below	168 (70.0)	72 (30.0)	
College and above	525 (73.7)	187 (26.3)	
Family average monthly income (Chinese yuan (CNY))			0.032
<5000	303 (70.8)	125 (29.2)	
5000–10,000	272 (77.7)	78 (22.3)	
>10,000	114 (68.3)	53 (31.7)	
Number of family members			0.052
3	233 (68.9)	105 (31.1)	
3–5	374 (73.9)	132 (26.1)	
>5	89 (80.2)	22 (19.8)	
Parity			0.613
1	458 (72.4)	175 (27.6)	
>2	235 (73.9)	83 (26.1)	
Delivery mode			0.479
Vaginal delivery	278 (71.6)	110 (28.4)	
Cesarean delivery	418 (73.7)	129 (26.3)	
Parity			
1	449(64.5)	247(35.5)	0.074
≥2	183(70.7)	76(29.3)	
Physical activities (MET hours/week)			0.515
<20	402 (73.5)	145 (26.5)	
≥20	282 (71.6)	112 (28.4)	
Self-reported sleep quality			<0.001
Good	210 (79.5)	54 (20.5)	
Fair	333 (78.0)	94 (22.0)	
Poor	135 (61.4)	85 (38.6)	
Very poor	18 (40.9)	26 (59.1)	
BMI			0.841
<18.9	38 (73.1)	14 (26.9)	
18.9–23.9	395 (72.2)	152 (27.8)	
≥24	259 (74.0)	91 (26.0)	

**Table 2 nutrients-14-00903-t002:** Postpartum depression among women with traditional postpartum practices or not (*n*, %).

	Postpartum Depression					
	Yes	No	*p*	OR (95% CI)	AOR ^a^ (95% CI)	AOR ^b^ (95% CI)
Overall postpartum practices						
No	359 (76.5)	110 (23.5)	0.012	Ref.	Ref.	Ref.
Yes	337 (69.3)	149 (30.7)		1.44 (1.08, 1.92)	1.44 (1.08, 1.93)	1.41 (1.04, 1.90)
Non-dietary practices						
No	508 (74.4)	175 (25.6)	0.099	Ref.	Ref.	Ref.
Yes	188 (69.1)	84 (30.9)		1.30 (0.95, 1.77)	1.35 (0.98, 1.84)	1.26 (0.92, 1.75)
Dietary practices						
No	459 (75.5)	149 (24.5)	0.016	Ref.	Ref.	Ref.
Yes	237 (68.3)	110 (31.7)		1.43 (1.07, 1.92)	1.41 (1.05, 1.90)	1.37 (1.01, 1.85)

^a^ Adjusting for age, family income, lactation stages, and number of family members. ^b^ Adjusting for age, family income, lactation stages, number of family members, and self-reported sleep quality.

**Table 3 nutrients-14-00903-t003:** The association between PPD-related dietary pattern score and traditional postpartum practices.

	Postpartum Depression-Related Dietary Pattern			
Dietary Customs	Mean ± SD	*p*	OR (95% CI)	AOR ^a^ (95% CI)
No	0.08 ± 0.99	0.001	Ref.	Ref.
Yes	0.14 ± 1.01		1.25 (1.10, 1.43)	1.25 (1.10, 1.43)

^a^ Adjusting for age, family income, lactation stage, and number of family members.

## Data Availability

The data presented in this study are available on request from the corresponding author. The data are not publicly available due to ethical requirements.

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
