# Peer review of "The Association between Postpartum Practice and Chinese Postpartum Depression: Identification of a Postpartum Depression-Related Dietary Pattern"

_nutrients, 2022, doi:10.3390/nu14040903_

Round 1

Reviewer 1 Report

Overall, the topic is interesting and the manuscript is well written. The statistics power seems not sufficient due to the study design. There are many inherent limitations in the study. Authors may want to discuss more potential limitations and interpret the results with cautions. Notably, information on some important risk factors of postpartum depression were not collected in this study, which may confound the result, e.g., the mother’s history of depression, the children’s health status, and number of children, etc. In addition, the data collection regarding self-reported sleep quality is vague and confusing. More discussion is needed. 

Author Response

We sincerely appreciate for the reviewer’s comment.

According to the suggestions, we carefully revised the paper. All the corrections and modifications we made were responded to the reviewers point by point.

  1. The statistics power seems not sufficient due to the study design.

Answer:We agree with the reviewer’s comment. Although the samples used from a previous study, the sample size was also calculated for the current analysis. According to the significance test of difference, we estimate the prevalence of postpartum depression in women with postpartum customs or not were 30% and 20% respectively, and the estimated sample size was around 600. However, we agree with the reviewer’ s comment on the sample size, The non-randomized sample method and a relatively small sample size may greatly undermine the reliability of the current results. We added the limitation and the research expectations in revised manuscript(limitation part).

2.There are many inherent limitations in the study. Authors may want to discuss more potential limitations and interpret the results with cautions. Notably, information on some important risk factors of postpartum depression were not collected in this study, which may confound the result, e.g., the mother’s history of depression, the children’s health status, and number of children, etc.

Answer:We sincerely thanks for the reviewer to point these limitations. Some of the confounders actually collected in this study but we did not well take into consideration and didn’t show the data in previous manuscript (we collected the parity). We added this information in revised manuscript. Some of the important information was not been collected and as the reviewer mentioned before, due to the study design, the potential immeasurable confounder may exist. We added this limitation in revised manuscript. All the words were revised with cautions.

  1. In addition, the data collection regarding self-reported sleep quality is vague and confusing. More discussion is needed. 

Answer: We appreciate the reviewer to point out this limitation. Indeed, the sleep quality was assessed in current study only according to the participants’ self-report. Although in this study we more focus on the subjective feeling of participants, since this study revealed a crucial role of sleep quality in developing depression, a more detailed measurement of sleep (including sleep hours and quality) measured with a validated questionnaire or more accurate tools is highly needed. We further discussed this in the revised manuscript.

.

Reviewer 2 Report

The topic is interesting. The article has some important aspects in the methodology that must be clarified and addressed. In the operationalization of the variables, it should be done in another way. The way it is currently can lead to errors. The time elapsed since delivery is important. The authors dichotomized into women who had given birth on more than 42 days or less than 42 days. It is evident that a woman who gave birth a year ago (365 days) is not the same as one who gave birth a little over a month ago (42 days). The Edinburgh Postnatal Depression Scale is indicated to identify the risk of postpartum depression at 6-8 weeks postpartum. The authors use it at 2 weeks and even at 50 weeks after delivery. The Edinburg Postnatal Depression Scale is used in a way that is not recommended. Food intake is collected through a frequency questionnaire. There is abundant literature that questions the validity of this instrument to be able to establish dietary patterns. For example, there is self-reported information and this influences the quality of the information and possible biases. Also, there is no approach to the limitations and biases and their possible influence on the results. For example, a confounding bias, an anamnestic bias, a classification bias, etc. The sleep variable is included. The role of this variable in an article on dietary patterns and its influence on the risk of postpartum depression is not well understood.

Author Response

We sincerely appreciate for the reviewer’s comments. It is a great help for us to improve the quality of the current study.

According to the suggestions, we carefully revised the paper. All the corrections and modifications we made were responded to the reviewers point by point.

  1. The time elapsed since delivery is important. The authors dichotomized into women who had given birth on more than 42 days or less than 42 days. It is evident that a woman who gave birth a year ago (365 days) is not the same as one who gave birth a little over a month ago (42 days). The Edinburgh Postnatal Depression Scale is indicated to identify the risk of postpartum depression at 6-8 weeks postpartum.

Answer:Thanks for reviewer proving this comment. We totally agree with the reviewers’ comment. Indeed, not only the postpartum depression, but the postpartum customs may greatly different between women in 1-2 months postpartum and after. Unfortunately, we didn’t collect the detailed data on the exact days after birth. Considering the depression highly prevalent in 1-2 months and the postpartum customs more common in 1-2 months after labor, we used the date 42 days as the cut-off, which the data (< 42days, or later) could be found in YI study. For the EPDS,it is recommend to apply for evaluating the depression in early stage of lactation,but it is also widely used in other postpartum stage,like 6 m, or even 1 year (some of the references were shown in below) .

According to the reviewers’ comment, we further adjusted the lactation stage in multi-variate model (unfortunately, we could only use 42days as the cut-off), the results are consistent with previous results (the results parts were revised).

Finally, although we did our best try to take the time issue into consideration, we agree with the reviewer, the influence by time elapsed after labor cannot be ignored. We added this significant limitation in the revised paper.

The references of the long term usage of EPDS

1.Smith-Nielsen, J., Matthey, S., Lange, T. et al. Validation of the Edinburgh Postnatal Depression Scale against both DSM-5 and ICD-10 diagnostic criteria for depression. BMC Psychiatry 18, 393 (2018). https://doi.org/10.1186/s12888-018-1965-7

2.Smith-Nielsen, J., Matthey, S., Lange, T. et al.Validation of the Edinburgh Postnatal Depression Scale against both DSM-5 and ICD-10 diagnostic criteria for depression. BMC Psychiatry18, 393 (2018). https://doi.org/10.1186/s12888-018-1965-7

3.Areias ME, Kumar R, Barros H, Figueiredo E. Comparative incidence of depression in women and men, during pregnancy and after childbirth. Validation of the Edinburgh Postnatal Depression Scale in Portuguese mothers. Br J Psychiatry. 1996 Jul;169(1):30-5. doi: 10.1192/bjp.169.1.30. PMID: 8818365.

  1. Food intake is collected through a frequency questionnaire. There is abundant literature that questions the validity of this instrument to be able to establish dietary patterns. For example, there is self-reported information and this influences the quality of the information and possible biases.

Answer: We agree with the food intake frequency questionnaire is not the best way to draw the dietary pattern. The three days 24 dietary record were more recommended. However, with a retrospective design, only food intake frequencies questionnaire and one day 24 dietary recall were used. And we believe a method which could reflect a relatively mid-long term dietary intake is more suitable for drawing dietary pattern, comparing with one time 24-hour recall. However, as the reviewer mentioned the bias could not be ignored, and we indeed not addressed this issue. We added this information in the limitation part.

  1. The sleep variable is included. The role of this variable in an article on dietary patterns and its influence on the risk of postpartum depression is not well understood.

Answer: We are sincerely thanks for the reviewer pointing out a really important issue. Actually, the role of sleep was identified in pathway analysis, which is out of our expectation. And based on the current literatures, the sleep both associated with postpartum customs and depression, but the possible pathway is not well understood. In previous version of paper, we did not well discuss the role of sleep. In addition, only sleep quality with self-report was used in the current study. To more clearly interpret this result, we rewrote the discussion part, and also list the limitation of evaluation of sleep quality in limitation part.